# Inclusion of persons living with disabilities in a district-wide sanitation programme: A cross-sectional study in rural Malawi

**Katherine Davies**[1]*, **Mindy Panulo**[2], **Clara MacLeod**[1], **Jane Wilbur**[4], **Tracy Morse**[3], **Kondwani Chidziwisano**[2], **Robert Dreibelbis**[1]

**1** Department of Disease Control, London School of Hygiene & Tropical Medicine (LSHTM), London, United Kingdom, **2** Centre for Water, Sanitation, Health, and Appropriate Technology Development (WASHTED), Malawi University of Business and Applied Sciences, Blantyre, Malawi, **3** Department of Civil and Environmental Engineering, University of Strathclyde, Glasgow, United Kingdom, **4** Department of Population Health, London School of Hygiene & Tropical Medicine (LSHTM), London, United Kingdom

* katherine.davies1@lshtm.ac.uk

**Data Availability Statement:** Data has been deposited in a repository (DOI: https://doi.org/10.17037/DATA.00003783).

## Abstract

Community-led total sanitation (CLTS) is embraced as a key strategy to achieve universal sanitation coverage (Sustainable Development Goal 6.2). Although inclusion is identified as a predictor of CLTS success, people living with disabilities are often excluded from community sanitation programmes and there is limited research exploring CLTS participation amongst people living with disabilities. This study aims to explore the extent to which people living with disabilities participated in a CLTS intervention delivered in rural Malawi using standard approaches. This cross-sectional study was conducted in the Chiradzulu district of Malawi. A household questionnaire was administered to collect information about CLTS participation. Multivariable logistic regression was performed to compare participation in different CLTS activities between households with (n = 80) and without a member with a disability (n = 167), and between household members with (n = 55) and without a disability (n = 226). No difference in CLTS participation was observed at the household-level, but there were marked differences in CLTS participation between household members with and without a disability. Household members without a disability felt they could give more input in triggering activities (OR = 3.72, 95%CI 1.18–11.73), and reported higher participation in the transect walk (OR = 4.03, 95%CI 1.45–11.18), community action planning (OR = 2.89, 95%CI 1.36–6.13), and follow-up visits (OR = 3.37, 95%CI 1.78–6.40) compared to household members with disabilities. There was no difference in the likelihood of being invited to triggering (OR = 0.98, 95%CI 0.41–2.36), attending triggering (OR = 2.09, 95%CI 0.98–4.46), or participating in community mapping (OR = 2.38, 95%CI 0.71–7.98) between household members with and without a disability. This study revealed intra-household inequalities in CLTS participation. To improve participation in CLTS interventions, facilitators should be trained on action steps to make CLTS more inclusive. Further research could include an in-depth analysis of predictors of CLTS participation amongst people living with disabilities, including disability types, severity and age.

**Funding:** Funding was awarded to RD by World Vision USA (grant number: 102239IC). The funder had no role in study design, data collection and analysis, decision to publish, or preparation of the manuscript. Travel for KD was supported by LSHTM MSc Project Funding.

**Competing interests:** The authors have declared that no competing interests exist.

## Introduction

Sustainable Development Goal (SDG) 6.2 aims to end open defecation and provide access to improved sanitation to all by 2030 paying special attention to those in vulnerable situations, including people living with disabilities [1]. People living with disabilities are described as individuals with long-term impairments which hinder their ability to participate fully and equally in society [2]. The World Health Organisation's (WHO) International Classification of Functioning, Disability and Health (ICF) framework provides a bio-psycho-social model of disability, viewing disablement as a consequence of external factors that limit one's ability to participate fully in society, in addition to personal impairments [3]. The literature highlights physical, social, institutional, and personal barriers that make accessing sanitation facilities more challenging for people living with disabilities [4–10]. Barriers to sanitation access can include infrastructural factors (uneven terrain, high steps or narrow entrances), discrimination and exclusion from participating in community sanitation meetings. The barriers to accessing sanitation mean people living with disabilities often lose their autonomy, resulting in a reported lack of privacy and dignity [4–6, 11]. Inaccessible sanitation facilities are a particular challenge for people living with disabilities who menstruate and people with incontinence who report limiting social interactions due to fear of discrimination or violence [11–15].

Studies in multiple countries have documented disparities in sanitation access between people with and without disabilities [16–19]. Reported barriers to sanitation access are most marked for people with more complex impairments and where people living with disabilities and their households are among the poorest groups of the population [16–19]. However, the disparity in sanitation access is specific to the individual. Multiple studies have documented little to no differences in sanitation access comparing households with and without a member living with a disability [16–19]. Traditional household-level measures of sanitation access, therefore, may mask important disparities in sanitation access within the household [20].

Community-led total sanitation (CLTS) is a behaviour change approach that aims to catalyse collective community action to adopt sanitation and end open defecation in rural settings [21]. It has been widely used in low-income countries as a key strategy to meet the Sustainable Development Goal sanitation targets as it is low-cost and can be used to rapidly reach large populations [22]. The focal activity of CLTS is the triggering session, where community members are 'triggered' by feelings of shame and disgust to critically self-assess their sanitation behaviours. The triggering session involves a transect walk (also known as a walk of shame) and a community mapping activity (Fig 1). The triggering session can lead to a commitment to end open defecation and the development of a community action plan to construct latrines using available local resources [21]. Post-triggering activities include household visits to discuss and monitor latrine construction progress against community action plans.

There is mixed evidence on CLTS success. While several studies report increased latrine coverage in communities following CLTS [23–26], a systematic review and meta-analysis revealed no statistically significant increase in latrine coverage or usage in households following CLTS interventions [27]. For CLTS to be successful and sustainable, there must be broad and inclusive community participation [28–30]. Excluding vulnerable community members, including people living with disabilities, can lead to these members returning to open defecation behaviours due to the construction of inaccessible latrines [31]. To ensure the design and location of sanitation facilities are accessible and usable, people living with disabilities must participate fully across the entire CLTS process [7].

People living with disabilities are often excluded from participating fully in community programmes, including sanitation programmes [32–34]. Socially excluded groups, including people living with disabilities, are inadequately involved in the design and planning of sanitation

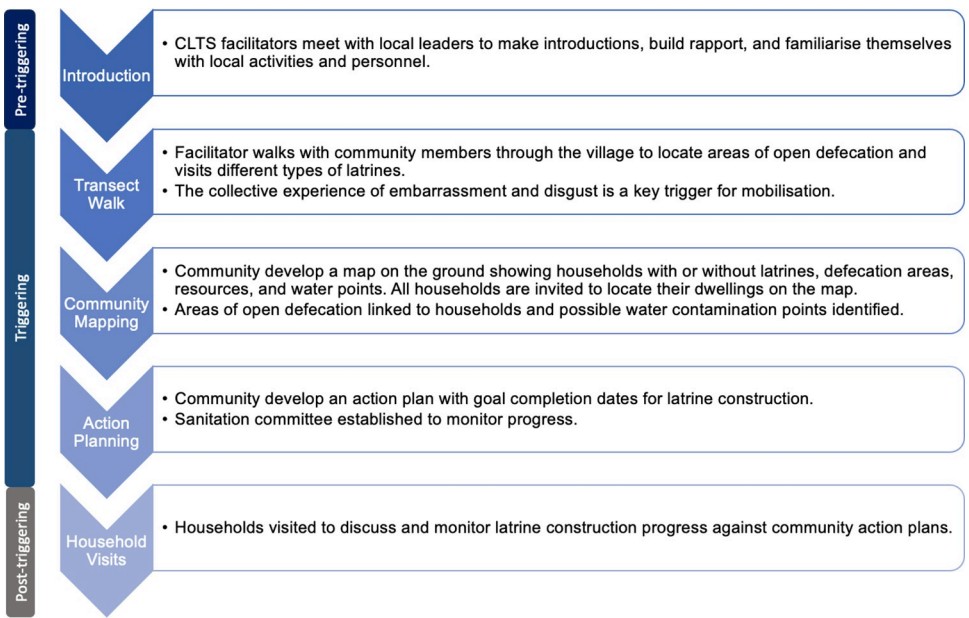

**Fig 1. Description of key CLTS activities.**

projects and frequently feel they are not listened to [35]. Physical barriers to participation for people living with disabilities include inaccessible meeting locations or a lack of provision of assistance (such as visual aids or verbal descriptions), whilst social barriers include stigma and discrimination, leading to the exclusion of people living with disabilities from community events [34, 36]. Communities often under-estimate the ability of people living with disabilities to participate in community meetings [37]. This can lead to people living with disabilities feeling inferior or rejected by the community, so they do not feel they can participate in community events. The inclusion of people living with disabilities in community programmes involves action by facilitators to ensure they are able to participate, to raise community awareness of the barriers to accessing sanitation and to disseminate information about low-cost adaptations to improve the accessibility of latrines such as ramps, wide entrances, handrails, and moveable seats [36, 38, 39].

Achieving universal sanitation coverage, as outlined in the sustainable development goal targets, requires addressing the needs of people living with disabilities, which is dependent on their full and equitable participation in sanitation programmes. People living with disabilities constitute upwards of 16% of the global population [40, 41]. Therefore, more information is needed to better understand their participation in community-based interventions. This exploratory study aims to address this gap by exploring the extent to which adults with disabilities (over the age of 18) participatied in a CLTS intervention delivered in rural Malawi. Specifically, the study compared CLTS participation between households with and without a member with a disability, and between household members with and without a disability.

## Methods

### Study setting

This cross-sectional study was conducted in Malawi. According to the 2018 Population and Housing Census, there are 1,734,250 people aged 5 years or older with at least one type of disability in Malawi, representing about 11.6% of the total population [42]. This includes people

with limitations in at least one functional domain (seeing, hearing, walking, speaking, intellectual, or self-care) as well as people with Albinism and Epilepsy. According to WHO/UNICEF Joint Monitoring Programme (JMP) estimates from 2022, around 50% of the population in Malawi lacks access to improved sanitation facilities and an estimated 531,000 people (2.61%) practice open defecation, which is largely concentrated in rural, poor areas [43]. This study focused on two Traditional Authorities (TAs) in the Chiradzulu district, situated in the Southern Region of Malawi. The district has a population of approximately 350,000 people and National Statistics Office data from 2018 estimates approximately 11% of the population have a disability [42]. Sanitation coverage in Chiradzulu has historically been below the national average. According to the 2018 Chiradzulu District Sector Investment Plan (DSIP), 53% (440/831) of villages were certified open defecation free (ODF) [44]. However, only 5.9% (49/831) of villages were certified ODF by the National ODF Task Force.

This study was conducted as part of a larger research and learning collaboration between the London School of Hygiene and Tropical Medicine (LSHTM), the Malawi University of Business and Applied Sciences (MUBAS), and the international non-governmental organisation (NGO) World Vision focusing on water, sanitation and hygiene (WASH) programming in Chiradzulu District, Malawi. Since 2022, World Vision Malawi, in partnership with the Malawi Government through Chiradzulu district council, has implemented the WASH for Everyone project, which aims to reach universal sanitation coverage in the district by the end of 2025. This study focused on the TAs of Mpama and Likoswe where WASH for Everyone partners completed CLTS activities in June 2022. The CLTS intervention delivered by the WASH for Everyone programme was delivered using standard approaches in accordance with national guidelines. The implementation did not have a specific focus on inclusions of persons with disabilities. Of note, a large tropical cyclone impacted the region between intervention implementation and data collection, affecting overall latrine coverage in the region.

## Sampling procedure

Multiple sampling approaches were used in the recruitment of study participants (Fig 2). We aimed to recruit a total of 250 households; 200 households using random sampling and 50 households using purposive sampling. This study was embedded within a larger process evaluation of World Vision's sanitation programmes in TAs Mpama and Likoswe with an estimated sample size of approximately 200 households. Simple random sampling was used to randomly select 10 villages from each TA and 10 households per selected village. To explore CLTS participation among people living with disabilities, we aimed to enroll an additional 50 households with a member living with a disability. A list of 25 households with a member living with a disability from TA Mpama and TA Likoswe was provided by the Chiradzulu district council social welfare officer. Purposive sampling was used to ensure an adequate number of households with a member living with a disability were interviewed within the study timeframe in addition to the 200 households randomly sampled as part of the wider process evaluation. Organisations of persons with disabilities (OPDs) are established only at the regional level so there was no formal disability structure in the local context. Therefore, we worked closely with the district council social welfare office, which considers the needs of people living with disabilities, through community health workers (CHWs), and social welfare extension officers to identify households and recruit study participants.

## Data collection

Questionnaires, consisting primarily of closed-ended questions with pre-coded responses, were developed by the study team and translated into the local language (Chichewa). Surveys

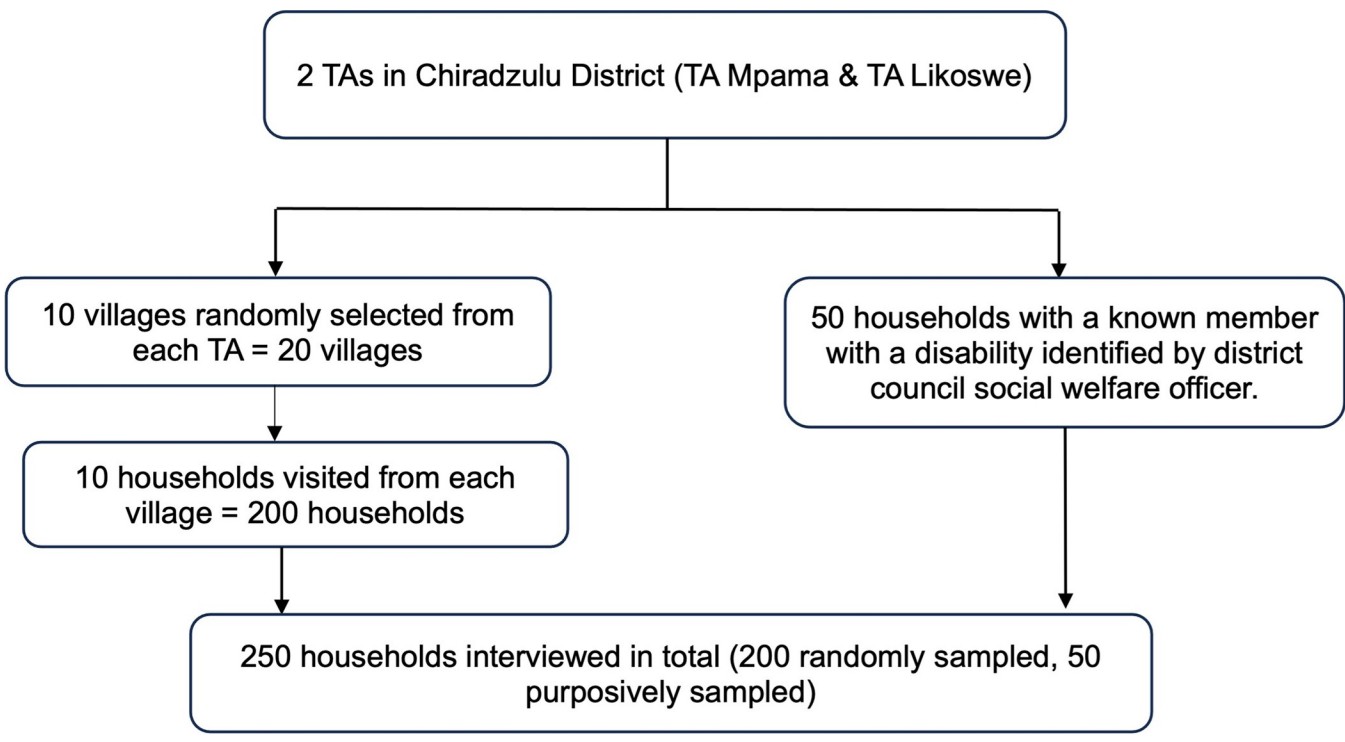

**Fig 2. Flow diagram outlining the sampling procedure.**

were piloted in two villages in Chiradzulu district that were not part of the final study sample. A team of four enumerators fluent in Chichewa were recruited for data collection activities. All enumerators had previous experience administering survey questions around sanitation. Prior to data collection, enumerators completed training on questionnaire content, data collection procedures and ethical safeguarding (informed consent, data protection, and disability awareness training).

Participants were recruited and data were collected between 26 June to 14 July 2023. Surveys were orally administered. Electronic data entry forms were built using Kobo Collect and administered on Android tablets. Data from the tablets were uploaded into a secure, cloud-based server daily. Prior to the survey being administered, an information sheet was read out by the interviewer outlining the study purpose and procedures before informed consent was obtained. No respondents under 18 were interviewed in this study and individuals with cognitive difficulties, where it was not possible to ascertain informed consent under field-based survey conditions, were not asked to participate.

In each participating household, surveys were first completed with the respondent identified as most responsible for water and sanitation in the household. (Fig 3). To determine if anyone in the household had a disability, the Washington Group Short Set (WG-SS) of questions on functioning were then administered with the primary household respondent [45]. Anyone in the household reported as having 'a lot of difficulty' or 'cannot do at all' in any one of the WG-SS functional domains (vision, hearing, communication, cognition, mobility, and self-care) was defined as having a disability. If a household member with a disability was present, the same survey was then completed with them. In most households where two participants were interviewed, the second respondent was the household member living with a disability. However, if the primary respondent was the household member living with a

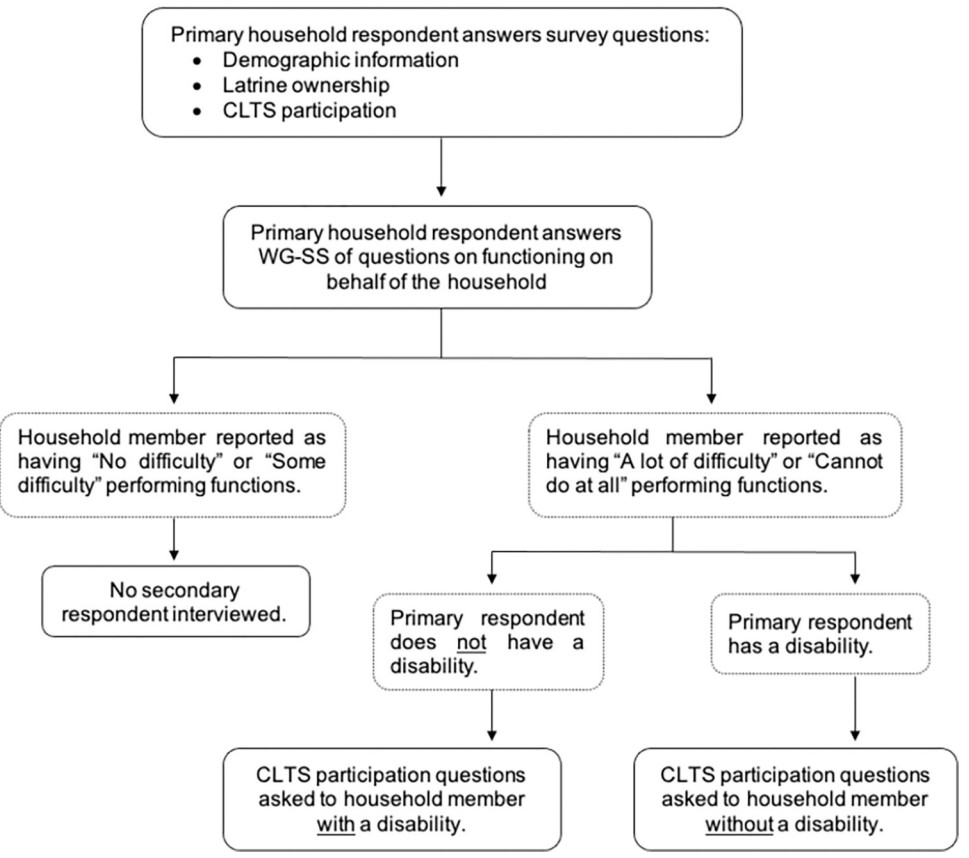

**Fig 3. Flow diagram outlining data collection procedures.**

disability, the second respondent recruited from the household was an adult household member without a disability. If the second respondent was not available or not eligible as per the exclusion criteria, only the primary respondent was interviewed but the household was still noted as having a member living with a disability.

Surveys with the primary household respondent captured household composition and sociodemographic information as well as latrine access and ownership. All respondents were asked a series of yes or no questions to assess participation in CLTS activities. Respondents were asked if they were invited to triggering, if they attended triggering, if their household was visited, and if they participated in key CLTS activities such as the transect walk, community mapping, and community action planning. Respondents were then asked to report if key activities to improve CLTS inclusivity were conducted by facilitators [38]. Specifically, they were asked if assistance was provided for people living with disabilities, if a squatting exercise was conducted, and if information on low-cost cost technologies to improve latrine accessibility was provided.

## Data management and analysis

Quantitative data from household questionnaires were exported from Kobo Collect to Microsoft Excel for data cleaning, then uploaded to STATA V17.0 for analysis. Analysis was conducted at both the household and individual level. At the household level, responses from the primary household respondents were used to determine household participation. Multiple

logistic regression was performed to calculate odds ratios to compare participation in selected elements of CLTS programming between households with and without a member living with a disability. Analyses were adjusted for gender of the primary respondent, income, education, household size and reported time since latrine construction. Analyses were also adjusted for potential clustering at the village level using cluster robust standard errors.

At the individual-level, multiple logistic regression models estimated odds ratios comparing CLTS programming participation between household members with and without a disability. Individual-level analyses were adjusted for age, sex, and income. Analyses were also adjusted for clustering of individuals within the same village using cluster robust standard errors. This level of clustering accounted for clustering of individuals within households. Models with interaction terms for gender were fitted to test if gender was an effect modifier in the relationship between disability and CLTS participation. The significance of effect modification was determined using a p-value cut-off of <0.05. Cross tabulations were performed to calculate the proportion of respondents reporting that activities important for inclusive CLTS were conducted by facilitators. Multiple logistic regression was performed to calculate odds ratios to compare reporting between household members with and without a disability with adjustment for age, sex, and income. Analyses were adjusted for clustering at the village level using cluster robust standard errors.

## Ethical considerations

Ethical approval was gained from LSHTM's MSc Research Ethics Committee (Ref 28569) and Malawi's National Committee on Research in the Social Sciences and Humanities (NCST/RTT/2/6, P.09/22/673). Additional information regarding the ethical, cultural, and scientific considerations specific to inclusivity in global research is included in the supporting information (S3 Text).

Informed consent was obtained from all participants interviewed and was confirmed by a thumb-print or written signature, depending on literacy status. Where the individual consenting was illiterate, a literate impartial witness signed to confirm that the participant understood and consented to the survey. Individuals with cognitive difficulties, where it was not possible to ascertain informed consent under field-based survey conditions, were not asked to participate. This was up to the discretion of the data collection team who worked closely with caregivers to decide whether participation in the study was appropriate. If it was felt the purpose of the study was not understood by the respondent at any stage, the interview was stopped. If the respondent had a visual impairment and they could not read or write, an impartial witness signed to confirm the participant understood and consented to the survey. If the respondent had a hearing impairment, the information sheets were given to the respondent to read and consent form signed as usual. If the respondent had a hearing impairment and was not literate, the respondent was not asked to participate unless they could sufficiently communicate. Despite efforts to use a sign language interpreter, we were unable to secure participation. We did not rely on family members to interpret due to risks of misinterpretation.

## Results

### Respondent characteristics

A total of 247 households were enrolled, of which 32% (80/247) had a member living with a disability. This included the 200 households randomly sampled and an additional 47 households purposively sampled, just short of our targeted purposive sample of 50 households. Seventeen percent (33/200) of households from the random sample had a member with a disability. Most of the lead household respondents interviewed were female (87%; 215/247),

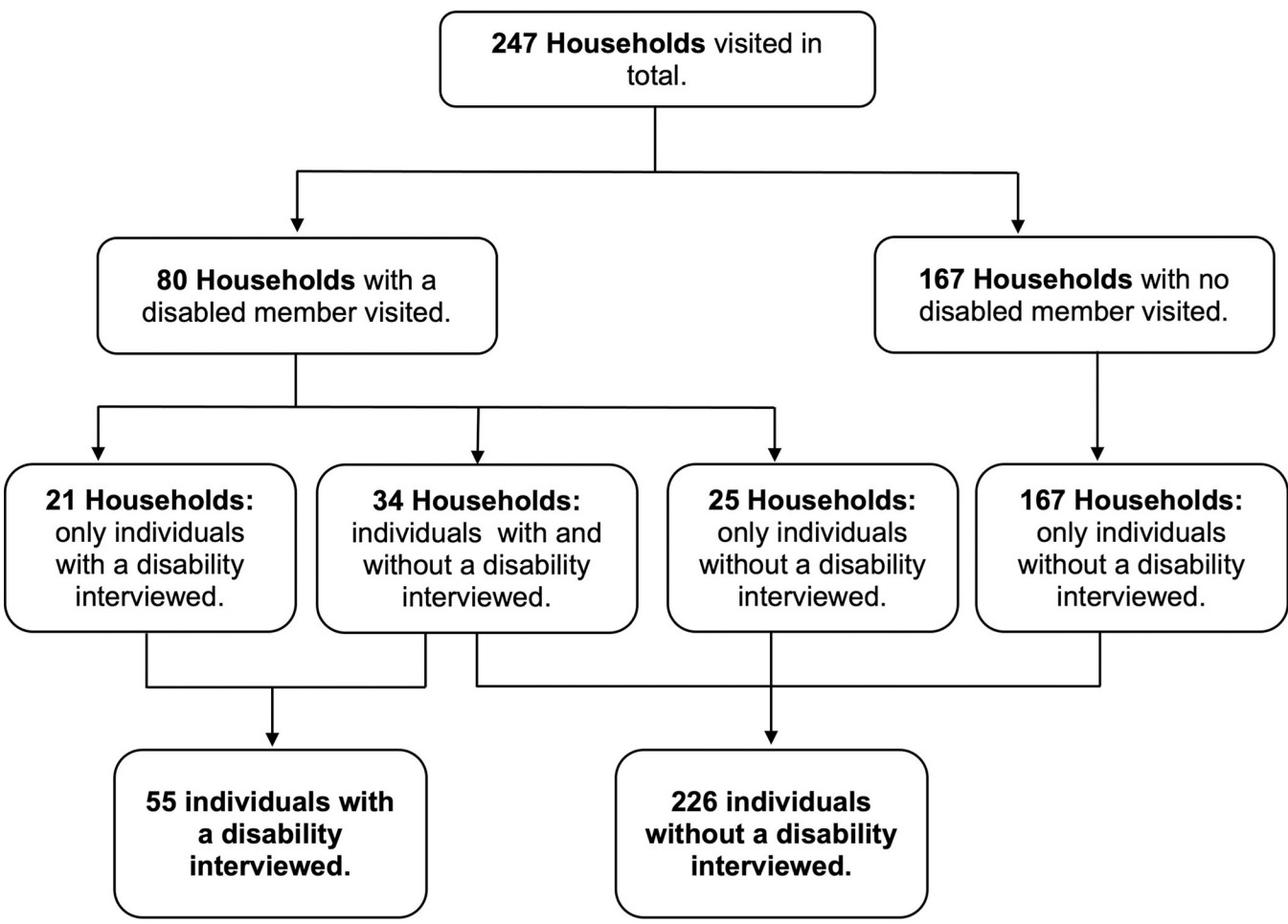

**Fig 4. Flow diagram outlining how the number of individuals interviewed was reached based on the number of households visited.**

married (59%; 145/247), and did not have a disability (90%; 222/247). Pit latrines were the only types of latrines owned by households, with more households owning pit latrines without a slab (53%; 131/247) than with a slab (9%; 22/247). Thirty-eight percent (94/247) of respondents reported that they did not own a latrine, with most of these households instead using a neighbour's latrine.

A total of 281 individuals were interviewed in the 247 households visited (Fig 4). Approximately 20% (55/281) of individuals interviewed had a disability. Most of the respondents were female (83%; 232/281). Respondents who did not have a disability were more likely to be female (92%; 208/226) than male (8%; 18/226). The gender distribution was more even among respondents with a disability, who were more likely to be male (56%; 31/55) than female (44%; 24/55). Mobility was the most common functional limitation reported (85%; 47/55) followed by self-care (38%; 21/55). Detailed sample characteristics are outlined in Tables 1 and 2.

## CLTS participation

**Household-level analysis.** Sixty-four percent (159/247) of primary respondents reported they were invited to the triggering session and roughly half (55%; 136/247) reported they attended the session. Reported participation in the transect walk (18%; 44/247) and

**Table 1. Household-level characteristics stratified by households with and without a member with a disability (n = 247).**

| Variable | Total (n = 247) | Households with a disabled member (n = 80) | Households with no disabled member (N = 167) |
|---|---|---|---|
| | N (%) | N (%) | N (%) |
| **Household Disability Status** | | | |
| Member with a disability | 80 (32%) | - | - |
| No member with a disability | 167 (68%) | - | - |
| **Lead Respondent Disability Status** | | | |
| Disability | 25 (10%) | 25 (31%) | 0 (0%) |
| No disability | 222 (90%) | 55 (69%) | 167 (100%) |
| **Lead Respondent Gender** | | | |
| Male | 32 (13%) | 21 (26%) | 11 (7%) |
| Female | 215 (87%) | 59 (74%) | 156 (93%) |
| **Lead Respondent Age** | | | |
| 18–24 years | 37 (15%) | 12 (15%) | 25 (15%) |
| 25–34 years | 62 (25%) | 11 (14%) | 51 (31%) |
| 35–44 years | 58 (23%) | 19 (24%) | 39 (23%) |
| 45–54 years | 31 (13%) | 12 (15%) | 19 (11%) |
| 55–64 years | 32 (13%) | 14 (18%) | 18 (11%) |
| 65+ years | 27 (11%) | 12 (15%) | 15 (9%) |
| **Household Size** | | | |
| 1–3 people | 89 (36%) | 26 (33%) | 63 (38%) |
| 3–5 people | 90 (36%) | 26 (33%) | 64 (38%) |
| 6+ people | 68 (28%) | 28 (35%) | 40 (24%) |
| **Household Monthly Income (Malawi Kwacha)** | | | |
| < 20,000 | 126 (51%) | 45 (56%) | 81 (49%) |
| 20,000–40,000 | 54 (22%) | 19 (24%) | 35 (21%) |
| 40,000 + | 67 (27%) | 16 (20%) | 51 (31%) |
| **Lead Respondent Marital Status** | | | |
| Single | 33 (13%) | 18 (23%) | 15 (9%) |
| Married | 145 (59%) | 43 (54%) | 102 (61%) |
| Divorced | 39 (16%) | 10 (13%) | 29 (17%) |
| Widowed | 30 (12%) | 9 (11%) | 21 (13%) |
| **Lead Respondent Education** | | | |
| Up to Primary | 187 (76%) | 66 (83%) | 121 (72%) |
| Secondary + | 60 (24%) | 14 (18%) | 46 (28%) |
| **Lead Respondent Occupation** | | | |
| Farming | 80 (32%) | 33 (41%) | 47 (28%) |
| Business | 51 (21%) | 14 (18%) | 37 (22%) |
| Piece works | 71 (29%) | 19 (24%) | 52 (31%) |
| Other | 45 (18%) | 14 (18%) | 31 (19%) |
| **Defecation Location** | | | |
| In dwelling toilet | 153 (62%) | 54 (68%) | 99 (59%) |
| Neighbours toilet | 92 (37%) | 24 (30%) | 68 (41%) |
| Open defecation | 2 (1%) | 2 (3%) | 0 (0%) |
| **Household Latrine Ownership** | | | |
| Own a latrine | 153 (62%) | 54 (68%) | 99 (59%) |
| Do not own a latrine | 94 (38%) | 26 (33%) | 68 (41%) |
| **Household Latrine Type** | | | |
| Pit latrine with a slab | 22 (9%) | 11 (14%) | 11 (7%) |

*(Continued)*

**Table 1.** (Continued)

| Variable | Total (n = 247) | Households with a disabled member (n = 80) | Households with no disabled member (N = 167) |
|---|---|---|---|
| | N (%) | N (%) | N (%) |
| Pit latrine without a slab | 131 (53%) | 43 (54%) | 88 (53%) |
| No latrine | 94 (38%) | 26 (33%) | 68 (41%) |
| **Household Latrine Construction Time** | | | |
| Before W4E project | 136 (55%) | 50 (63%) | 86 (52%) |
| After W4E project | 17 (7%) | 4 (5%) | 13 (8%) |
| No latrine | 94 (38%) | 26 (33%) | 68 (41%) |
| **Functional Limitation Type (Disabled Household Member)** * | | | |
| Vision | - | 4 (5%) | - |
| Hearing | - | 9 (11%) | - |
| Mobility | - | 55 (69%) | - |
| Cognition | - | 5 (6%) | - |
| Self-care | - | 30 (38%) | - |
| Communication | - | 11 (14%) | - |

*Domains are not mutually exclusive

**Table 2. Individual-level characteristics stratified by individuals with and without a disability (n = 281).**

| Variable | Total (n = 281) | Individuals with disabilities (n = 55) | Individuals without disabilities (n = 226) |
|---|---|---|---|
| | N (%) | N (%) | N (%) |
| **Disability Status** | | | |
| Disability: | 55 (20%) | - | - |
| No disability | 226 (80%) | - | - |
| **Gender** | | | |
| Male | 49 (17%) | 31 (56%) | 18 (8%) |
| Female | 232 (83%) | 24 (44%) | 208 (92%) |
| **Age** | | | |
| 18–24 | 41 (15%) | 6 (11%) | 35 (15%) |
| 25–34 | 66 (23%) | 10 (18%) | 56 (25%) |
| 35–44 | 64 (23%) | 8 (15%) | 56 (25%) |
| 45–54 | 36 (13%) | 8 (15%) | 28 (12%) |
| 55–64 | 39 (14%) | 11 (20%) | 28 (12%) |
| 65+ | 35 (12%) | 12 (22%) | 23 (10%) |
| **Monthly Income (Malawi Kwacha)** | | | |
| < 20,000 | 147 (52%) | 35 (64%) | 112 (50%) |
| 20,000–40,000 | 61 (22%) | 10 (18%) | 51 (23%) |
| 40,000 + | 73 (26%) | 10 (18%) | 63 (28%) |
| **Functional Limitation Type*** | | | |
| Vision | - | 2 (4%) | - |
| Hearing | - | 3 (5%) | - |
| Mobility | - | 47 (85%) | - |
| Cognition | - | 0 (0%) | - |
| Self-care | - | 21(38%) | - |
| Communication | - | 2 (4%) | - |

*Domains are not mutually exclusive

**Table 3. Adjusted and unadjusted odds ratios for the association between having a household member with a disability and participation in each element of CLTS (n = 247).**

| CLTS Activity | Participation | | Unadjusted Odds Ratios | | Adjusted Odds Ratios | |
|---|---|---|---|---|---|---|
| | N/Total | % | OR | 95% CI | aOR[†] | 95% CI |
| **Invited to triggering** | | | | | | |
| Household member with a disability | 51/80 | 64% | 1.00 | - | 1.00 | - |
| No household member with a disability | 108/167 | 65% | 1.04 | 0.60–1.81 | 0.68 | 0.38–1.24 |
| **Total** | **159/247** | **64%** | - | - | - | - |
| **Attended triggering** | | | | | | |
| Household Member with a disability | 40/80 | 50% | 1.00 | - | 1.00 | - |
| No household member with a disability | 96/167 | 57% | 1.35 | 0.79–2.31 | 1.03 | 0.57–1.85 |
| **Total** | **136/247** | **55%** | - | - | - | - |
| **Participated in transect walk** | | | | | | |
| Household Member with a disability | 15/80 | 19% | 1.00 | - | 1.00 | - |
| No household member with a disability | 29/167 | 17% | 0.91 | 0.46–1.81 | 0.76 | 0.38–1.50 |
| **Total** | **44/247** | **18%** | - | - | - | - |
| **Participated in community mapping** | | | | | | |
| Household Member with a disability | 11/80 | 14% | 1.00 | - | 1.00 | - |
| No household member with a disability | 30/167 | 18% | 1.37 | 0.65–2.90 | 1.15 | 0.54–2.42 |
| **Total** | **41/247** | **17%** | - | - | - | - |
| **Felt they could give input** | | | | | | |
| Household Member with a disability | 21/80 | 26% | 1.00 | - | 1.00 | - |
| No household member with a disability | 57/167 | 34% | 1.46 | 0.81–2.63 | 1.65 | 0.88–3.10 |
| **Total** | **78/247** | **32%** | - | - | - | - |
| **Participated in community action planning** | | | | | | |
| Household Member with a disability | 28/80 | 35% | 1.00 | - | 1.00 | - |
| No household member with a disability | 65/167 | 39% | 1.18 | 0.68–2.06 | 1.08 | 0.61–1.90 |
| **Total** | **93/247** | **38%** | - | - | - | - |
| **Visited by someone to discuss latrine construction/use** | | | | | | |
| Household Member with a disability | 59/80 | 74% | 1.00 | - | 1.00 | - |
| No household member with a disability | 134/167 | 80% | 1.45 | 0.77–2.71 | 1.53 | 0.79–2.99 |
| **Total** | **193/247** | **78%** | - | - | - | - |

*P<0.05 in the Wald Test

[†]Adjusted for gender of the primary respondent, income, education, household size and reported time since latrine construction, with consideration for clustering at the village level.

community mapping (17%; 41/247) was lower; while over three-quarters of lead respondents (78%; 193/247) reported their household was visited by someone to discuss latrine contruction or usage after CLTS events.

While participation in CLTS amongst primary households repsondents was proportionally lower in households with a member living with a disability for most activities, there was no statistical evidence that having a household member with a disability decreased the likelihood of participating in any CLTS activities (Table 3) (Tables A-G in S1 Text).

**Individual-level analysis.** Participation in CLTS was proportionally lower among household members living with a disability for all activities. Individual-level regression models found no association between having a disability and being invited to triggering (OR = 0.98, 95%CI 0.41–2.36), attending triggering (OR = 2.09, 95%CI 0.98–4.46) or participating in community mapping (OR = 2.38, 95%CI 0.71–7.98). However, household members without a

disability had greater odds of participating in all other aspects of CLTS programming. Household members without a disability gave more input in triggering activities (OR = 3.72, 95%CI 1.18–11.73), and reported higher participation in the transect walk (OR = 4.03, 95%CI 1.45–11.18), community action planning (OR = 2.89, 95%CI 1.36–6.13), and follow-up visits (OR = 3.37, 95%CI 1.78–6.40) compared to household members living with a disability (Table 4) (Tables A-G in S2 Text). The data did not support the hypothesis that gender was an effect modifier of the relationship between disability and CLTS participation (S1 Table).

**CLTS inclusivity.** Further indicators of interest were investigated to assess the inclusivity of CLTS sessions delivered. Of the 148 individuals who attended the triggering sessions, 20% (29/148) reported that assistance was provided to support the participation of individuals living with a disability and 14% (21/148) reported that a squatting demonstration was given. Only 7% (21/281) of the respondents reported they were provided with information on low-cost technologies to improve access to WASH for people living with disabilities. There were minimal differences in reporting between individuals with and without a disability (Table 5).

**Table 4. Adjusted and unadjusted odds ratios for the association between having a disability and participation in each element of CLTS (n = 281).**

| CLTS Activity | Participation | | Unadjusted Odds Ratios | | Adjusted Odds Ratios | |
|---|---|---|---|---|---|---|
| | N/Total | % | OR | 95% CI | aOR[†] | 95% CI |
| **Invited to triggering** | | | | | | |
| Individuals with disabilities | 32/55 | 58% | 1.00 | - | 1.00 | - |
| Individuals without disabilities | 147/226 | 65% | 1.34 | 0.73–2.44 | 0.98 | 0.41–2.36 |
| **Total** | **179/281** | **64%** | - | - | - | - |
| **Attended triggering** | | | | | | |
| Individuals with disabilities | 19/55 | 35% | 1.00 | - | 1.00 | - |
| Individuals without disabilities | 129/226 | 57% | 2.52** | 1.36–4.66 | 2.09 | 0.98–4.46 |
| **Total** | **148/281** | **53%** | - | - | - | - |
| **Participated in transect walk** | | | | | | |
| Individuals with disabilities | 4/55 | 7% | 1.00 | - | 1.00 | - |
| Individuals without disabilities | 42/226 | 19% | 2.91* | 1.00–8.50 | 4.03** | 1.45–11.18 |
| **Total** | **46/281** | **16%** | - | - | - | - |
| **Participated in community mapping** | | | | | | |
| Individuals with disabilities | 3/55 | 5% | 1.00 | - | 1.00 | - |
| Individuals without disabilities | 42/226 | 19% | 3.96* | 1.18–13.28 | 2.38 | 0.71–7.98 |
| **Total** | **45/281** | **16%** | - | - | - | - |
| **Felt they could give input** | | | | | | |
| Individuals with disabilities | 7/55 | 13% | 1.00 | - | 1.00 | - |
| Individuals without disabilities | 76/226 | 34% | 3.47** | 1.50–8.04 | 3.72* | 1.18–11.73 |
| **Total** | **83/281** | **30%** | - | - | - | - |
| **Participated in community action planning** | | | | | | |
| Individuals with disabilities | 11/55 | 20% | 1.00 | - | 1.00 | - |
| Individuals without disabilities | 92/226 | 41% | 2.75** | 1.35–5.59 | 2.89** | 1.36–6.13 |
| **Total** | **103/281** | **37%** | - | - | - | - |
| **Visited by someone to discuss latrine construction/use** | | | | | | |
| Individuals with disabilities | 35/55 | 64% | 1.00 | - | 1.00 | - |
| Individuals without disabilities | 179/226 | 79% | 2.18* | 1.15–4.11 | 3.37** | 1.78–6.40 |
| **Total** | **214/281** | **76%** | - | - | - | - |

*P<0.05 in the Wald Test

**P<0.01 in the Wald Test

[†] Adjusted for age, sex and income with consideration for clustering at the village level.

Table 5.  Reported indicators of CLTS inclusivity stratified by individuals with and without a disability (n = 281).

| CLTS Inclusivity Indicator | Participation | | Unadjusted Odds Ratios | | Adjusted Odds Ratios | |
|---|---|---|---|---|---|---|
| | N/Total | % | OR | 95% CI | aOR^c | 95% CI |
| **Assistance provided for individuals with a disability** | | | | | | |
| Individuals with disabilities † | 4/19 | 21% | 1.00 | - | 1.00 | - |
| Individuals without disabilities † | 25/129 | 19% | 0.90 | 0.28–2.87 | 1.25 | 0.30–5.23 |
| **Total Individuals †** | **29/148** | **20%** | - | - | - | - |
| **Squatting demonstration given** | | | | | | |
| Individuals with disabilities † | 3/19 | 16% | 1.00 | - | 1.00 | - |
| Individuals without disabilities † | 18/129 | 14% | 0.86 | 0.23–3.19 | 1.09 | 0.25–4.83 |
| **Total Individuals †** | **21/148** | **14%** | - | - | - | - |
| **Low-cost technologies for accessibility provided** | | | | | | |
| Individuals with disabilities | 3/55 | 5% | 1.00 | - | 1.00 | - |
| Individuals without disabilities | 18/226 | 8% | 1.50 | 0.42–5.35 | 1.55 | 0.44–5.44 |
| **Total** | **21/281** | **7%** | - | - | - | - |

*P<0.05 in the Wald Test

† n = 148 (individuals who attended triggering only)

** Adjusted for age, sex and income with consideration for clustering at the village level.

## Discussion

While participation in CLTS was proportionally lower in households with a member living with a disability for most CLTS activities, there was no statistical evidence that having a household member with a disability decreased the likelihood of participating in any CLTS activities. In constrast, household members living with a disability were far less likely to participate in key CLTS activities compared to household members without a disability.

Important intrahousehold inequalities in CLTS participation were not captured using household-level measures in this analysis. These findings are consistent with the literature, which found minimal differences in sanitation access at the household-level, but reported differences in sanitation access between individuals with and without a disability [16–19]. Intrahousehold inequalities in sanitation access also exist amongst groups on the basis of gender, age, ethnicity and other social identities [46, 47]. As current global WASH indicators largely rely on national household surveys, with data usually collected from the head of household, it is likely these intrahousehold inequalities are not effectively captured [48]. The literature finds heads of household can over-report sanitation access for people with disabilities [20]. Therefore, interviewing individuals is important to reliably capture the needs of vulnerable and marginalised groups [48].

The findings from this study indicate the facilitation of standard CLTS interventions are not sufficiently inclusive to support the attendance and participation of people with disabilities. Only 20% of individuals reported that assistance was provided to support the participation of individuals living with a disability during triggering. Furthermore, only 7% of respondents reported that they were provided with information on low-cost technologies to make latrines more comfortable and safer for people with disabilities. A lack of information regarding latrine accessibility has been identified as a barrier to latrine access for people living with disabilities [49]. Therefore, it is important that existing resources outlining low-cost technologies to make sanitation facilities more accessible are widely shared during CLTS interventions [39, 49]. In this intervention, information on low-cost technology options could have been provided after triggering during household visits [36], especially as over 70% of households reported that they were visited to discuss latrine construction and usage.

The literature exploring participation in community programmes identifies a number of physical, social, institutional, and personal barriers to participation for people living with disabilities [4, 34, 36, 37, 50]. The limited capacity of facilitators to deliver inclusive CLTS programmes can contribute to the reported barriers to CLTS participation [7, 33, 37]. Facilitators in the WASH for Everyone intervention were not specifically trained to deliver inclusive CLTS. CLTS guidelines lack specific sections addressing the needs of people living with disabilities [21]. Furthermore, national strategies and data collection tools used for ODF verification do not adequately consider disability-related factors [51, 52]. Capacity development of government bodies and stakeholders is necessary to ensure they can effectively facilitate the involvement of people living with disabilities in CLTS activities [36, 37]. Training facilitators on action steps to make CLTS more inclusive can improve the inclusivity of CLTS interventions delivered and promote higher participation of people living with disabilities [53, 54]. However, limited improvements to latrine accessibility have been observed following inclusive CLTS, with latrine safety and accessibility remaining sub-optimal for people living with disabilities 18-months after an inclusive CLTS intervention [55]. Despite this, the inclusive CLTS approach represents an important step in promoting the inclusion of people living with disabilities in CLTS activities.

## Study limitations

Purposive sampling techniques required for this study mean results may not be generalizable beyond populations with similar characteristics as those included in our study. It is likely the district council social welfare officer knew of households where someone had a visible disability. Therefore, possible respondents may have been missed. Furthermore, we did not include individuals with cognitive disabilities, so their experiences are not included. As the disability assessment questions were answered by the lead household member and not validated by the individuals with disabilities, there could be misclassification of disability status as household heads commonly under-report disability status [56]. Finally, the results could be affected by recall bias if recall of CLTS participation differs between people with or without a disability. However, the binary nature of the CLTS participation questions means this is unlikely.

As this study was quantitative in nature, some key findings could not be evaluated or explored in further detail. Therefore, conducting further qualitative studies would allow aspects such as barriers to participation to be researched in further depth. Further research using a larger and more representative sample of the disabled population would improve the generalisability of the research to the wider population. This research could include more people with cognitive disabilities and children. Partnering with OPDs and interpreters would be important to support disability inclusion. Our analysis exploring effect modification did not find evidence of interaction between gender and disability. However, we note the limited number of male respondents who did not have a disability. These results should be interpreted with caution. Future studies that are adequately powered to explore the combined effects of gender and disability are needed. A larger sample size would also enable a more in-depth analysis of the predictors of CLTS participation amongst people living with a disability, including disability types, disability severity, and age. This research would be useful to ascertain why participation is lower amongst people living with disabilities.

## Conclusion

This study aimed to fill a gap in the sanitation and disability literature by investigating the participation of people living with disabilities in a CLTS intervention in rural Malawi. This cross-sectional study was conducted using household questionnaires to compare CLTS participation

between households with and without a member living with a disability and between house-hold members with and without a disability. Whilst no difference in CLTS participation was observed at the household-level, CLTS participation was lower for household members living with a disability compared to household members without a disability. Whilst there was no dif-ference in the likelihood of being invited to participate in CLTS, the likelihood of participating in key CLTS activities was lower amongst household members living with a disability.

Equitable participation in CLTS is crucial to the success and sustainability of CLTS out-comes. Therefore, to improve the participation of people living with disabilities, it is recom-mended that implementors should train facilitators on action steps to make CLTS more inclusive. While CLTS facilitators in the WASH for Everyone intervention were not specifi-cally trained to deliver inclusive CLTS; our findings demonstrate the potential shortcomings of standard approaches and suggest that deliberate action is needed to ensure true commu-nity-wide participation. The omission of disability-inclusive activities from CLTS guidelines impacts the participation of people living with disabilities in CLTS initiatives. Addressing this gap is essential for inclusive sanitation practices and ensuring no community member is left behind. Further research could use a larger and more representative sample to enable a more in-depth analysis of the predictors of CLTS participation or could include a qualitative analysis of the barriers to CLTS participation for people living with disabilities.

## Supporting information

**S1 Text. Full regression results for the association between having a household member with a disability and CLTS participation.**
(DOCX)

**S2 Text. Full regression results for the association between having a disability and CLTS participation.**
(DOCX)

**S3 Text. Inclusivity in global research questionnaire.**
(DOCX)

**S1 Table. P-values for interaction terms between gender and disability from fully adjusted regression models examining the association between disability and participation in each element of CLTS.**
(XLSX)

## Author Contributions

**Conceptualization:** Katherine Davies, Mindy Panulo, Clara MacLeod, Tracy Morse, Kond-wani Chidziwisano, Robert Dreibelbis.

**Data curation:** Katherine Davies.

**Formal analysis:** Katherine Davies.

**Funding acquisition:** Robert Dreibelbis.

**Investigation:** Katherine Davies, Mindy Panulo.

**Methodology:** Katherine Davies, Mindy Panulo, Clara MacLeod, Jane Wilbur, Kondwani Chidziwisano, Robert Dreibelbis.

**Project administration:** Mindy Panulo, Clara MacLeod, Kondwani Chidziwisano.

**Resources:** Kondwani Chidziwisano.

**Supervision:** Clara MacLeod, Kondwani Chidziwisano, Robert Dreibelbis.

**Visualization:** Katherine Davies.

**Writing – original draft:** Katherine Davies.

**Writing – review & editing:** Katherine Davies, Mindy Panulo, Clara MacLeod, Jane Wilbur, Tracy Morse, Kondwani Chidziwisano, Robert Dreibelbis.

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
