## [Decision Letter · Decision Letter 0]

16 Apr 2024

PGPH-D-24-00333

Evaluation of a district-wide sanitation programme in rural Malawi: does it include people living with disabilities?

Dear Dr. Davies,

Thank you for submitting your manuscript to PLOS Global Public Health. After careful consideration, we feel that it has merit but does not fully meet PLOS Global Public Health’s publication criteria as it currently stands. Therefore, we invite you to submit a revised version of the manuscript that addresses the points raised during the review process. Specifically, the reviewers requested clarification on methods and sample sizes for recruitment, accounting for clustering in the data analysis and effect modification by gender.

We look forward to receiving your revised manuscript.

Kind regards,

Ayse Ercumen, Ph.D.

Academic Editor

Journal Requirements:

Additional Editor Comments (if provided):

Reviewers' comments:

Reviewer's Responses to Questions

**Comments to the Author**

1. Does this manuscript meet PLOS Global Public Health’s publication criteria? Is the manuscript technically sound, and do the data support the conclusions? The manuscript must describe methodologically and ethically rigorous research with conclusions that are appropriately drawn based on the data presented.

Reviewer #1: Yes

Reviewer #2: Yes

2. Has the statistical analysis been performed appropriately and rigorously?

Reviewer #1: Yes

Reviewer #2: Yes

3. Have the authors made all data underlying the findings in their manuscript fully available (please refer to the Data Availability Statement at the start of the manuscript PDF file)?

Reviewer #1: Yes

Reviewer #2: Yes

4. Is the manuscript presented in an intelligible fashion and written in standard English?

Reviewer #1: Yes

Reviewer #2: Yes

5. Review Comments to the Author

Reviewer #1: The authors report a cross-sectional study of community-led total sanitation (CLTS) which was conducted in the Chiradzulu district of Malawi and aimed to explore the extent to which people living with disabilities participated in a CLTS intervention delivered in rural Malawi. The study compared CLTS participation between households with and without a member with a disability,

and between household members with and without a disability.

I did not identify any major issues to be addressed, only a couple of minor points for editing.

1. page 12 line 213 states Washing Group Short Set (WG-SS) which I think should be Washington Group Short Set (WG-SS).

2. For reading clarity, I suggest format tables 1 and 2 to have the percentages (%) closer to relevant number within each column, as currently formatted they are closer to the numbers for the subsequent column.

Reviewer #2: Please see comments attached in a separate document.

Extra text added to meet minimum character limit. Extra text added to meet minimum character limit. Extra text added to meet minimum character limit.

6. PLOS authors have the option to publish the peer review history of their article (what does this mean?). If published, this will include your full peer review and any attached files.

**Do you want your identity to be public for this peer review?** For information about this choice, including consent withdrawal, please see our Privacy Policy.

Reviewer #1: **Yes: **Jo-Anne Lee Geere

Reviewer #2: No

---

## [Editor Report · Decision Letter 1]

29 Jul 2024

PGPH-D-24-00333R1

Inclusion of persons living with disabilities in a district-wide sanitation programme: a cross-sectional study in rural Malawi.

Dear Dr. Davies,

Thank you for submitting your manuscript to PLOS Global Public Health. After careful consideration, we feel that it has merit but does not fully meet PLOS Global Public Health’s publication criteria as it currently stands. Therefore, we invite you to submit a revised version of the manuscript that addresses the points raised during the review process. Specifically, please provide a full description of the methods used to assess effect modification by gender, and include numerical findings from this analysis (please see detailed comments below).

We look forward to receiving your revised manuscript.

Kind regards,

Ayse Ercumen, Ph.D.

Academic Editor

Journal Requirements:

Additional Editor Comments (if provided):

Thank you for addressing the comments from the peer reviewers. Before the paper can be accepted for publication, please also address the following:

1) Describe the methods used to assess effect modification by gender (subgroup analysis or use of interaction terms, if the latter, the p-value cut-off for defining significant effect modification) as well as the numerical results from this analysis. In the results section, please report the gender breakdown among individuals with vs. without disabilities in the text, and include ORs and p-values from the effect modification analysis in the text and/or in a table. Please also comment on how the small stratum of males without disabilities may have affected the effect modification analysis.

2) Please replace singular conjugations following data (e.g., "data was") with plural conjugations (e.g., "data were") throughout the text.
---

## [Editor Report · Decision Letter 2]

6 Aug 2024

Inclusion of persons living with disabilities in a district-wide sanitation programme: a cross-sectional study in rural Malawi.

PGPH-D-24-00333R2

Dear Dr. Davies,

We are pleased to inform you that your manuscript 'Inclusion of persons living with disabilities in a district-wide sanitation programme: a cross-sectional study in rural Malawi.' has been provisionally accepted for publication in PLOS Global Public Health.

Best regards,

Ayse Ercumen, Ph.D.

Academic Editor